# Oral Squamous Cell Carcinoma Cells with Acquired Resistance to Erlotinib Are Sensitive to Anti-Cancer Effect of Quercetin via Pyruvate Kinase M2 (PKM2)

**DOI:** 10.3390/cells12010179

**Published:** 2023-01-01

**Authors:** Chien-Yi Chan, Shih-Cing Hong, Chin-Ming Chang, Yuan-Hong Chen, Pin-Chen Liao, Chun-Yin Huang

**Affiliations:** 1Department of Nutrition and Health Sciences, Chang Jung Christian University, Tainan 711301, Taiwan; 2Department of Nutrition, China Medical University, Taichung 406040, Taiwan

**Keywords:** erlotinib, OSCC, PKM2, quercetin

## Abstract

Oral squamous cell carcinoma (OSCC) frequently carries high epidermal growth factor receptor (EGFR) expression. Erlotinib, a small molecule tyrosine kinase inhibitor (TKI), is an effective inhibitor of EGFR activity; however, resistance to this drug can occur, limiting therapeutic outcomes. Therefore, in the current study, we aimed to unveil key intracellular molecules and adjuvant reagents to overcome erlotinib resistance. First, two HSC-3-derived erlotinib-resistant cell lines, ERL-R5 and ERL-R10, were established; both exhibited relatively higher growth rates, glucose utilization, epithelial-mesenchymal transition (EMT), and invasiveness compared with parental cells. Cancer aggressiveness-related proteins, such as N-cadherin, Vimentin, Twist, MMP-2, MMP-9, and MMP-13, and the glycolytic enzymes PKM2 and GLUT1 were upregulated in ERL-R cells. Notably, ERL-R cells were sensitive to quercetin, a naturally-existing flavonol phytochemical with anti-cancer properties against various cancer cells. At a concentration of 5 μM, quercetin effectively arrested cell growth, reduced glucose utilization, and inhibited cellular invasiveness. An ERL-R5-derived xenograft mouse model confirmed the growth-inhibitory efficacy of quercetin. Additionally, knock-down of PKM2 by siRNA mimicked the effect of quercetin and re-sensitized ERL-R cells to erlotinib. Furthermore, adding quercetin blocked the development of erlotinib-mediated resistance by enhancing apoptosis. In conclusion, our data support the application of quercetin in anti-erlotinib-resistant OSCC and indicate that PKM2 is a determinant factor in erlotinib resistance and quercetin sensitivity.

## 1. Introduction

Oral squamous cell carcinoma (OSCC), often categorized as a head and neck squamous cell carcinoma (HNSCC), frequently carries a high level of epidermal growth factor (EGF) receptors (EGFRs), which is correlated with poor clinical outcome [1]. Erlotinib (Tarceva), a first-generation small molecule tyrosine kinase inhibitor (TKI), is an orally reversible inhibitor of the wild-type EGFR autophosphorylation which competes for ATP binding [2]. Although the use of erlotinib for patients with OSCC/HNSCC has not yet been approved by the U.S. Food and Drug Administration (FDA) [3], numerous related experimental and clinical studies continue to be carried out. A phase II study evaluating the safety and effectiveness of erlotinib in patients with advanced recurrent and/or metastatic HNSCC indicated that erlotinib was safe [4] and patient survival was improved [5]. In contrast, another trial showed no significant improvement in complete response rate (CRR); however, a trend toward improvement of progression-free survival (PFS) with the addition of erlotinib to standard cisplatin-radiotherapy treatment was observed [6]. However, acquired insensitivity to the drug can occur, contributing to its limited positive clinical outcomes [7]. Thus, finding intracellular molecules which contribute to, and adjuvant therapies to overcome erlotinib resistance appears to be necessary.

Quercetin (3,3′,4′,5,7-pentahydroxyflavone) is a flavonoid which occurs naturally in various fruits and vegetables, including onions and apples, as well as in red wine [8]. Clinical trials have indicated that quercetin is safe, and that its presence in plasma can effectively inhibit the activity of lymphatic tyrosine kinase [9]. Quercetin exhibits multiple biological activities such as anti-inflammatory, anti-oxidation, anti-proliferative and pro-apoptotic properties [8]. Additionally, it has also been shown to inhibit the aggressive progression of various cancers, including OSCC [10,11]. The synergistic effect of quercetin with various compounds on cancer therapy has been widely reported [12]; however, the effects of its application in drug-resistant cancer cells, especially OSCC, require elucidation.

Pyruvate kinase M2 (PKM2), one of the four isoforms of pyruvate kinase (PK) that typically catalyze the conversion of phosphoenolpyruvate (PEP) to pyruvate in the glycolysis process, is highly expressed in various cancer cells, including OSCC [13]. Its upregulation positively correlates with increased glucose utilization in cancer cells [14]. Upregulation of PKM2 has also been reported in cancer cells resistant to drugs such as cisplatin [15] and erlotinib [16]. However, its role in OSCC cells resistant to erlotinib remains unexplored.

In the current study, we aimed to investigate the potential application of quercetin in OSCC cells with acquired resistance to erlotinib and the role of PKM2 in conferring resistance to erlotinib and sensitivity to quercetin. Moreover, we also set out to clarify if adding quercetin would hinder the development of the erlotinib-resistant OSCC cells. Data from the present study might support the use of quercetin as an alternative therapeutic agent in treating OSCC cells resistant to erlotinib and may indicate PKM2 suppression as a rational approach to sensitizing otherwise erlotinib-resistant OSCC.

## 2. Materials and Methods

### 2.1. Reagents and Antibodies

Quercetin and other chemicals were purchased from Sigma (St. Louis, MO, USA) unless specified otherwise. Erlotinib (Tarceva, OSI Pharmaceuticals, Melville, NY, USA) was purchased from Lumtec (Hsinchu, Taiwan), antibodies against Akt, E-cadherin, GAPDH, p-Akt, and β-actin were purchased from Santa Cruz Biotechnology (Santa Cruz, CA, USA), and anti-p-EGFR antibodies were purchased from Millipore (Burlington, MA, USA). Anti-EGFR, ERK, HK2, LDHA, MMP-2, PKM2, p27, Twist, and Vimentin antibodies were purchased from GeneTex, Inc. (Irvine, CA, USA). Antibodies against GLUT1, p-ERK, N-cadherin, and α-tubulin were obtained from Cell Signaling Technology (Danvers, MA, USA). MMP-9 and p21 antibodies were obtained from Abcan (Cambridge, UK) and Proteintech (Rosemont, IL, USA), respectively. PKM2 siRNA and the Lipofectamine RNAiMAX reagent were purchased from Invitrogen (Grand Island, NY, USA).

### 2.2. Cell Culture and Treatment

The human OSCC cell lines HSC-3, OECM-1, and SAS were kindly gifted to us by Drs. Tzong-Ming Shieh at China Medical University (Taichung, Taiwan). The HSC-3 cell line was maintained in Dulbecco’s modified Eagle medium (DMEM)/F-12 supplemented with 10% fetal bovine serum (FBS) and 1% antibiotic-antimycotic and maintained in a 37 °C, 5% CO_2_ incubator. All cell culture reagents were obtained from Invitrogen (Carlsbad, CA, USA) unless otherwise indicated.

### 2.3. Establishment of Erlotinib-Resistant Cell Lines

To establish erlotinib-resistant cell lines, HSC-3 cells were continuously exposed to erlotinib for 6 months. The ERL-R5 line was created by incubating cells with media containing 3 µM erlotinib for 1 month followed by 5 µM erlotinib for another 5 months. The ERL-R10 line was created by exposing cells to 3 µM erlotinib for 1 month followed by 5 µM for 1 month and 10 µM for another 4 months. The medium was replaced twice a week and cells were passaged at 80% confluence.

### 2.4. Cell Viability Assay

A 3-(4,5-dimethylthiazol-2-yl)-2,5-diphenyltetrazolim bromide (MTT) colorimetric dye reduction assay (Sigma, St. Louis, MO, USA) was used to determine the cell viability. Cells were plated at 20,000 cells/well in a 24-well plate and allowed to grow for 24 h. After treatment with the indicated chemicals, cell media were replaced with 400 μL of 0.5 mg/mL MTT reagent for 3 h. The formazan formed in cells was dissolved in isopropanol by wrapping the plate in foil and shaking on an orbital shaker for 15 min. Absorbance was measured at OD = 570 nm using a microplate reader (Model 680, Bio-Rad, Hercules, CA, USA).

### 2.5. Colony Formation Assay

Subsequently, 2 × 10^3^ cells were seeded into 6-well plates and incubated with growth media containing the indicated concentration of erlotinib or quercetin for 7 days [17]. Then, cells were washed with PBS, fixed with 10% formalin for 10 min, and then stained with 0.5% crystal violet (Panreac Quimica S.A.U.) for 30 min. Cell plates were subjected to colony counting and optical density measurement at 540 nm.

### 2.6. Cell Cycle Analysis

Cells on 6-well plates after treatment for the indicated times were trypsinized, washed with phosphate-buffered saline, and fixed in cold 70% ethanol at −20 °C [18]. The fixed cells were then collected and stained in a solution containing 0.5 mL of 4 μg/mL of propidium iodide (PI), 0.5 mg/mL of RNase, and 1% Triton X-100 for 30 min at 4 °C. The cell cycle distribution was analyzed based on DNA content using a FACScan flow cytometer (Becton Dickinson, Franklin Lakes, NJ, USA).

### 2.7. Apoptosis Analysis

After treatment, cells in a 6-well plate were trypsinized, washed with PBS, and suspended in a 1× binding buffer from the Annexin V-FITC apoptosis detection kit (BD Biosciences Pharmingen, San Diego, CA, USA) [17]. Then, 100 μL of the cell solution was transferred to a polystyrene tube, and 5 μL of Annexin V-FITC and 5 μL of PI were added for staining. The tubes were sheltered with foil and incubated for 15 min at room temperature. Finally, 400 μL of 1× binding buffer was added to each tube, and cells were analyzed using a FACScan flow cytometer (Becton Dickinson).

### 2.8. Cell Migration and Invasion Assay

A 6.5 mm Transwell^®^ with 8.0 µm Pore Polycarbonate Membrane Inserts (Corning Incorporated, Corning, NY, USA) was used for the Transwell migration assay [18]. Matrigel inserts were purchased from BD Biosciences (Bedford, MA, USA) for invasion assay and operated according to the manufacturer’s protocol. Briefly, cell suspensions (1 × 10^5^ cells) with the indicated treatments were seeded to the upper chamber, and 10% FBS-containing medium was added to the lower chamber of a 24-well plate as a chemoattractant. After incubation in a 37 °C, 5% CO_2_ incubator, the non-migrated and non-invading cells on the upper surface of the Transwell or Matrigel inserts were removed using cotton swabs, and the invaded cells on the lower surface were fixed with 100% methanol and stained with Giemsa in 20% ethanol. The migrated and invaded cells were photographed and counted in five randomly selected microscopic fields. Error bars represent the variations of the cell numbers between the chosen fields.

### 2.9. Glucose Uptake Assay

Glucose uptake was measured using the glucose analog, 2-deoxy-2-[(7-nitro-2,1,3-benzoxadiazol-4-yl) amino]-D-glucose; 2-NBDG) (Cayman Chemical Company, Ann Arbor, MI, USA). Cells (2 × 10^4^/well) cultured in a 24-well plate were treated with quercetin (0, 5 μM) for 24 h and then replaced with a low-glucose medium (5.55 mM) for 4 h in an incubator. Then, 150 μL of 10 μM 2-NBDG medium was added to each well and incubated for 10 min, followed by washing with 1× PBS. Cells with 2-NBDG were photographed under a fluorescence microscope, and fluorescent intensity was measured and normalized with cell number.

### 2.10. Aggregation Formation Assay

First, 2 × 10^3^ cells were seeded into low adhesive 35-mm culture dishes and cultured in regular growth media with or without 5 μM quercetin at 37 °C under 5% (*v*/*v*) CO_2_ [19]. The aggregation of cells was observed on the indicated day and recorded using a phase-contrast microscope.

### 2.11. Three-Dimensional (3D) Matrigel Assay

A 3D sphere formation assay was performed according to the protocol described by Storz et al. [20]. Briefly, a 24-well plate was pre-coated with undiluted, phenol red-free Matrigel (10 mg/mL) and a total of 10^4^ cells per well in 200 µL of phosphate-buffered saline (PBS) mixed with 100 µL of cold Matrigel (10 mg/mL) was added to each well. Culture media with or without 5 µM quercetin were replaced every other day. Photos were taken on the indicated day with 400× magnification.

### 2.12. Xenograft Mice

The animal study protocol, CMUIACUC-2022-168, was reviewed and approved by the Institutional Animal Care and Use Committee (IACUC) of the China Medical University. Three-week-old male nude mice were purchased from National Laboratory Animal Center (Taipei, Taiwan) and given one-week for acclimation with free access to autoclaved food and water. Mice were then subcutaneously implanted with human erlotinib-resistant HSC-3 cells (ERL-R5) (1 × 10^6^) suspended in PBS with liquid Matrigel (2:1) (final volume of 200 μL) into the flank regions and randomly assigned into three groups, with six mice per group. Starting one week after implantation, quercetin (0, 2, 10 mg/kg/day) dissolved in 1% DMSO of PBS solution was given every day via intraperitoneal injection, and tumor volume was evaluated by the equation ½ (length × width^2^). After 18 days, mice were sacrificed, and tumors were weighed.

### 2.13. Statistical Analysis

Data are expressed as mean ± SD from at least three independent experiments. Significant differences among group were analyzed by one-way ANOVA followed by Tukey’s test. *p* value < 0.05 was considered statistically significant. Graphics were plotted with the GraphPad Prism 8.0 software.

## 3. Results

### 3.1. Establishment and Characterization of Acquired Erlotinib-Resistant OSCC Cell Lines

The HSC-3 human tongue squamous carcinoma cell line was chosen to establish the erlotinib-resistant cell lines, since it exhibited higher phosphorylated and total EGFR protein levels than the other tested OSCC cell lines, i.e., OECM-1 and SAS (Figure 1A), and was sensitive to erlotinib (Figure 1B). Therefore, HSC-3 cells were maintained in media with increasing concentrations of erlotinib for six months, as described in Materials and Methods. Accordingly, two erlotinib-resistant cell lines, ERL-R5 and ERL-R10, were obtained based on the growth inhibitory efficacy of 5 and 10 μM of erlotinib, respectively (Figure 1C). These two cell lines remained insensitive to erlotinib even after switching back to regular growth media for four days, as evidenced by the clonogenic survival assay (Figure 1D).

To characterize erlotinib-resistant HSC-3 cells, we first switched the culture media for the insensitive cells back to growth media for four days and compared their growth features with parental cells. Erlotinib-resistant ERL-R5 and ERL-R10 cells displayed elevated growth capacities (Figure 1E) and up-regulated p-Akt and p-ERK protein levels, yet the protein levels of p-EGFR, p21, and p27 remained suppressed (Figure 1F) as compared with parental HSC-3 cells. Notably, in ERL-R10 cells, adding 10 μM erlotinib to growth media for 24 h was able to suppress the protein levels of p-Akt and p-ERK (Figure 1G), yet the cell viability (Figure 1C) and the expression of two known erlotinib-regulated cell cycle inhibitors, p21 and p27, remained unchanged (Figure 1G). These data indicated that down-regulation of p21 and p27 might be crucial for the acquired insensitivity to erlotinib.

### 3.2. Acquired Erlotinib-Resistant Cells Were Sensitive to the Growth-Inhibitory Effect of Quercetin In Vitro and In Vivo

Quercetin is well-known for its anti-cancer properties. Previously, we reported that quercetin potently inhibited HSC-3 cell viability at an IC_50_ value of 20 μM/24 h [17]. Hence, we wondered if erlotinib-resistant HSC-3 cells would also be sensitive to quercetin. Surprisingly, erlotinib-resistant cells exhibited an even lower IC_50_ concentration (5 μM) than the parental HSC-3 (Figure 2A). Further, the clonogenic survival data confirmed the stronger inhibitory effect of quercetin on colony expansion in the erlotinib-resistant cells (Figure 2B). To understand the underlying mechanism responsible for the growth-inhibitory effect of quercetin, we performed a flow cytometry analysis of the cell cycle progression. The result revealed that 5 μM of quercetin led to increased cell accumulation in the S and G2 phases in ERL-R5 and in the G2 phase in ERL-R10 cells (Figure 2C). Furthermore, 5 μM of quercetin did not suppress the expression of the G2 phase-related proteins such as cyclin B1 and cyclin A (Figure 2D) but induced the expression of p27 and p21 (Figure 2E) in both erlotinib-resistant cell lines. Together, these data indicated that quercetin at 5 μM could cause cell growth arrest in G2, at least in part, through induction of p27 and p21 in erlotinib-resistant cells. The anti-cancer effect of quercetin on human erlotinib-resistant HSC-3 cells was further elucidated with a xenograft nude mouse model. Compared to the vehicle control group, smaller tumors were burdened in the two quercetin-treated groups (Figure 3A). Moreover, a significant inhibition in tumor growth was found in the group treated with 10 mg/kg/day of quercetin. (Figure 3B), while tumor weight was significantly lower in both quercetin groups than in the control (Figure 3C). Together, the in vitro and in vivo data indicated that erlotinib-resistant HSC-3 cells were sensitive to the growth-inhibitory effect of quercetin.

### 3.3. The Acquired Malignant Phenotype of Erlotinib-Resistant Cells Was Sensitive to Quercetin

Next, we examined the malignant phenotype of the resistant cells and the effect of quercetin. The results from cell migration and invasion assays indicated that erlotinib-resistant cells were more aggressive than parental HSC-3 cells (Figure 4A,B). Since one crucial step for cancer cells to become metastatic is to escape from extracellular matrix (ECM) detachment-caused cell death [21], we further examined this possibility in erlotinib-resistant cells. The aggregation formation assay revealed that erlotinib-resistant cells were more aggregated in a suspension culture than non-resistant cells, indicating that erlotinib-resistant cells were able to evade detachment-mediated apoptosis (Figure 4C). However, the administration of 5 μM quercetin potently suppressed these acquired invading and aggregating capacities (Figure 4A–C). In addition, resistant cells showed a more significant sphere formation with invading edge than the non-resistant control in the 3-D Matrigel culture assay, although 5 μM quercetin potently suppressed this phenomenon (Figure 4D). Moreover, erlotinib-resistant cells displayed down-regulated protein expression of E-cadherin, an epithelial biomarker, but up-regulated expression of multiple mesenchymal biomarkers, including N-cadherin, Vimentin, and Twist (Figure 4E). The administration of 5 μM quercetin caused a marked elevation of E-cadherin but a reduction of N-cadherin and Twist in both erlotinib-resistant cell lines (Figure 4E). Surprisingly, 5 μM quercetin did not significantly affect the Vimentin protein level. Moreover, erlotinib-resistant cells exhibited up-regulated MMP-2, 9, and 13, essential ECM-disrupting proteins, whereas quercetin (5 μM) was able to suppress the expression of these invasion-promoting proteins (Figure 4F). Together, these data indicated that erlotinib-resistant cells displayed more aggressive features than those of parental cells; however, quercetin could suppress the acquired phenotype.

### 3.4. Erlotinib-Resistant Cells Exhibited Active Glucose Utilization and Elevated PKM2 Expression

To explore if erlotinib resistance is related to glucose utilization, we compared the glucose uptake capacity and aerobic glycolysis between erlotinib-sensitive and insensitive cells. A 2-NBDG (2-(N-(7-Nitrobenz-2-oxa-1,3-diazol-4-yl) Amino)-2-Deoxyglucose) uptake assay revealed that resistant HSC-3 cells displayed higher glucose uptake than the non-resistant control, as observed by fluorescent microscopy and quantification of the intracellular fluorescent intensity (Figure 5A). Surprisingly, examination of the glucose uptake and glycolysis-related proteins indicated a potent elevation of glucose transporter protein type 1 (GLUT1) and pyruvate kinase M2 (PKM2) proteins in the resistant cells (Figure 5B). The administration of 5 μM quercetin significantly suppressed the glucose uptake of the erlotinib-resistant HSC-3 cells, as evidenced by the considerable reduction of intracellular 2-NBDG intensity (Figure 5A). Quercetin also prominently down-regulated GLUT1, PKM2, and lactate dehydrogenase A (LDHA) expression of erlotinib-resistant HSC-3 cells (Figure 5B). To gain a possible clinical connection of GLUT1 (encoded by the SLC2A1 gene) and PKM2 (encoded by the PKM2 gene) to OSCC, we assessed both gene expression in the head and neck squamous cell carcinoma (HNSCC) database of the Cancer Genome Atlas (TCGA) project through the UALCAN web portal [22]. Our TCGA data analysis revealed that the mean expressions of GLUT1 and PKM2 in the primary tumors (*n* = 520) were significantly higher than that of normal tissue (*n* = 44) (Appendix A). Still, only higher PKM2 expression was significantly associated with a lower overall survival probability (Appendix A). Since PKM2 has also been reported to promote HIF-1α-mediated activation of glucose metabolism-related genes, including GLUT1, in cancer cells [23], it was rational to hypothesize that PKM2 might play a pivotal role in the acquired resistance to erlotinib and aggressive phenotype of OSCC. As predicted, the administration of PKM2 siRNA resulted in decreased cell viability (Figure 5C) and invasion (Figure 5D) in a dose-dependent manner. Additionally, suppression of PKM2 reduced the expression of GLUT1, N-cadherin, Twist, MMP-2, MMP-9, and NNP13 but induced the expression of E-cadherin, p21, and p27 in the ERL-R5 cell line (Figure 5E). Moreover, a co-expression analysis of a subset of the HNSCC database (530 samples) of the TCGA project using the cBioPortal website revealed a significant positive correlation between PKM2 and GLUT1, HK2, LDHA, MMP-2, MMP-9, and MMP-13 (Appendix A) [24]. These data indicated that PKM2 was a key regulator of glucose utilization, cell growth, and cell invasiveness, and that knock-down of PKM2 would mimic the effect of quercetin on erlotinib-resistant cells.

### 3.5. Knock-Down of PKM2 Re-Sensitized Resistant Cells to Erlotinib

If PKM2 contributed to the acquisition of resistance to erlotinib, it would be reasonable to hypothesize that knock-down of PKM2 would re-sensitize resistant cells to erlotinib. Indeed, the co-administration of erlotinib and PKM2 siRNA did provide additional inhibition of cell viability (Figure 6A) and invasion (Figure 6B) in resistant cell lines ERL-R5 and ERL-R10. In addition, co-administration resulted in a further reduction of N-cadherin, Twist, MMP-9, and MMP-13 protein levels but an enhanced induction of E-cadherin and p27 (Figure 6C,D). The addition of erlotinib also enhanced PKM2 knockdown-induced apoptosis, as evidenced by an increase in the positive Annexin V-PI staining cell numbers (Figure 6E) and cleaved-caspase 3 protein level (Figure 6F). Collectively, lowering PKM2 might assist resistant cells in regaining sensitivity to erlotinib.

### 3.6. Quercetin Suppressed the Development of Erlotinib Resistance through Apoptosis

Finally, we wondered if co-administration of quercetin and erlotinib would prevent the development of erlotinib resistance. During the first two weeks of co-treatment, quercetin seemed to suppress the development of resistance (Figure 7A). Indeed, a short-term (24 h) addition of quercetin exhibited a synergistic growth-inhibitory effect with erlotinib in a dose-dependent manner (Figure 7B). The co-administration of 5 μM quercetin and erlotinib resulted in additional inhibition of cell invasion compared to erlotinib alone; however, the co-administration of 10 μM quercetin and erlotinib did not lead to a further reduction of the invading capacity (Figure 7C). Unexpectedly, quercetin did not promote but somehow weakened the effect of erlotinib on the expression of proteins involved in the regulation of cell growth, EMT program, and invasion (Figure 7D). Nonetheless, the addition of quercetin enhanced the pro-apoptotic effect of erlotinib, as demonstrated by increased positive Annexin V-PI staining cell numbers (Figure 7E) and cleaved-caspase 3 expression (Figure 7F). This observation might explain the dose-responsive effect of quercetin on cell viability upon co-administration with erlotinib in HSC-3 cells (Figure 7A). Thus, these data suggested that quercetin may suppress the development of erlotinib resistance by enhancing apoptosis in HSC-3 cells.

## 4. Discussion

In the present study, we demonstrated that quercetin at 5 µM caused significant cell growth arrest, reduced glucose utilization, and blunted invasive progression in erlotinib-resistant HSC-3 OSCC cells. This concentration is low when compared with those used in other in vitro studies. Maximum apoptosis was observed in LNCaP (30.64%), PC-3 (27.9%), and DU-145 (27.2%) prostate cancer cells after a 72-h treatment with 40 μM of quercetin [25]. After 24 h of incubation, the same dose of quercetin differentially reduced cell viability by between 20% and 47% in nine different cancer cell lines (colon carcinoma CT-26 cells, prostate adenocarcinoma LNCaP cells, human prostate PC3 cells, pheocromocytoma PC12 cells, estrogen receptor-positive breast cancer MCF-7 cells, acute lymphoblastic leukemia MOLT-4 T-cells, human myeloma U266B1 cells, human lymphoid Raji cells, and ovarian cancer CHO cells) [26]. The IC_50_ for colon cancer cell lines HCT-15 and RKO were 142.7 and 121.9 μM/24 h, respectively [27]. After 5 days of treatment, the IC_50_ values of quercetin ranged from 44 to 105 μM for breast cancer cell lines MCF-7, BT549, HBL100, and MDA-MB-231, and 21 to 240 μM for ovarian cancer cell lines OVCAR3, TOV112D, OVCAR5, and CAOV3 [28]. Moreover, quercetin (30 μM) suppressed glycolysis in the MCF-7 and MDA-MB-231 breast cancer cells, as evidenced by decreased glucose uptake and lactate production with a concomitant decrease in the levels of the GLUT1, PKM2, and LDHA proteins [29]. Hexokinase 2 (HK2)-mediated glycolysis was also shown to be inhibited following quercetin treatment (25~50 μM) in Bel-7402 and SMMC-7721 hepatocellular carcinoma (HCC) cells [30]. Taken as a whole, our in vitro data support the use of quercetin as an effective alternative therapeutic reagent against erlotinib-resistant OSCC cells.

Identifying crucial intracellular molecules implicated in erlotinib insensitivity would help in choosing proper therapeutic strategies. Recently, upregulation of PKM2 was reported in erlotinib-resistant lung cancer cells; mechanistically, this might be due to a decrease in long non-coding (lnc) RNA H19-mediated, ubiquitin-induced PKM2 degradation, such that PKM2 confers resistance to erlotinib through phosphorylation of AKT [16]. Similarly, we also observed elevated PKM2 protein levels in two erlotinib-resistant HSC-3 cell lines (Figure 5B) with no change in PKM2 mRNA content (data not shown). Notably, knock-down of PKM2 by introducing PKM2-specific siRNA reduced the expression levels of the EMT and invasion-related proteins (Figure 5E), which was consistent with a previous finding in tongue squamous cell carcinoma (TSCC) [13]. Moreover, PKM2 may suppress the expression of E-cadherin by recruiting histone deacetylase 3 (HDAC3) to the promoter of E-cadherin gene, which requires interaction between PKM2 and TGF-β-induced factor homeobox 2 (TGIF2) [14]. Downregulation of PKM2 also potently restored sensitivity to the inhibitory effect of erlotinib on cell growth and invasion (Figure 6). In summary, upregulation of PKM2 expression appears to be crucial in acquiring resistance to erlotinib.

The p27 protein, a cyclin-dependent kinase (CDK) inhibitor and a negative regulator of cell cycle control, is a crucial downstream effector of the EGFR tyrosine kinase inhibitor (TKI)-mediated growth blockage in cancer cells [31,32]. Previously, we demonstrated that p27 protein was induced in erlotinib-mediated growth arrest in HSC-3 OSCC cells [18]. It was also reported that erlotinib-induced p27 phosphorylation at Serine 10 (S10) led to its cytoplasmic localization that conferred cell resistance to erlotinib [32]. However, how the p27 level is related to erlotinib resistance remains unclear. Here, we found that erlotinib resistant HSC-3 cells carried elevated PKM2 but reduced p27 protein levels, and that knock-down of PKM2 resulted in increased p27 expression. These findings prompted us to link PKM2 to p27 expression. A study by Mukherjee et al. clarified that PKM2 knockdown might lead to increased cap-independent translation of p27 mRNA by disrupting its interaction with HuR, an RNA-binding protein, without affecting p27 gene transcription, which was associated with cell accumulation in the G2 phase in glioblastoma [33]. In agreement with these findings, in erlotinib-resistant HSC-3 cells, knock-down of PKM2 also led to the upregulation of p27 and decreased cell viability (Figure 5C,E), which mimicked the effect of quercetin (Figure 2C,E), Additionally, knock-down of PKM2 re-sensitized resistant cells to the induction of p27 by erlotinib (Figure 6C,D). Apparently, the PKM2/p27 axis plays a decisive role in acquired erlotinib resistance and the anti-cancer effect of quercetin.

Quercetin displays a synergistic efficacy when used in combination with several anti-cancer drugs. The combined use of erlotinib and quercetin was shown to be more effective against A549 and NCI H460 cells than combinations of erlotinib with fisetin, carnosic acid, or luteolin [34]. In addition, erlotinib-quercetin nanoparticles significantly inhibited the expression of p-EGFR/PI3K/p-AKT protein in erlotinib-resistant A549/ER lung cancer cells in vitro and in vivo [34]. In agreement with these findings, here, we showed that the addition of quercetin promoted the inhibitory effect of erlotinib on cell viability and delayed the development of erlotinib-resistant cells by enhancing apoptosis (Figure 7A,B,E,F). However, adding quercetin to erlotinib did not exhibit a synergistic effect on the expression of some proteins involved in the EMT and invasion; on the contrary, this counteracted the effect of erlotinib (Figure 7D). This phenomenon might be due to cells’ fight-back mechanism against apoptosis, although we do not have direct evidence for this. The limitation of the current study is the indetermination of the combination index between quercetin and erlotinib. We cannot exclude the possibility of an additive or synergistic effect in other concentration ranges. In this regard, future in vitro and in vivo studies with various dose combinations of erlotinib and quercetin might solve the puzzle.

## 5. Conclusions

In conclusion, as depicted in Figure 8, oral cancer cells with acquired resistance to erlotinib exhibited high glucose utilization rate, cell growth, epithelial-mesenchymal transition program, and invading capacity. These resistant cells were sensitive to quercetin. PKM2 appeared to be a crucial factor in the development of erlotinib-resistant cells, and PKM2 knockdown mimicked the anti-cancer effect of quercetin and re-sensitized resistant cells to erlotinib. Adding quercetin to erlotinib blunted the initiation of the resistant cells, at least partially, through the induction of apoptosis.

## Figures and Tables

**Figure 1 cells-12-00179-f001:**
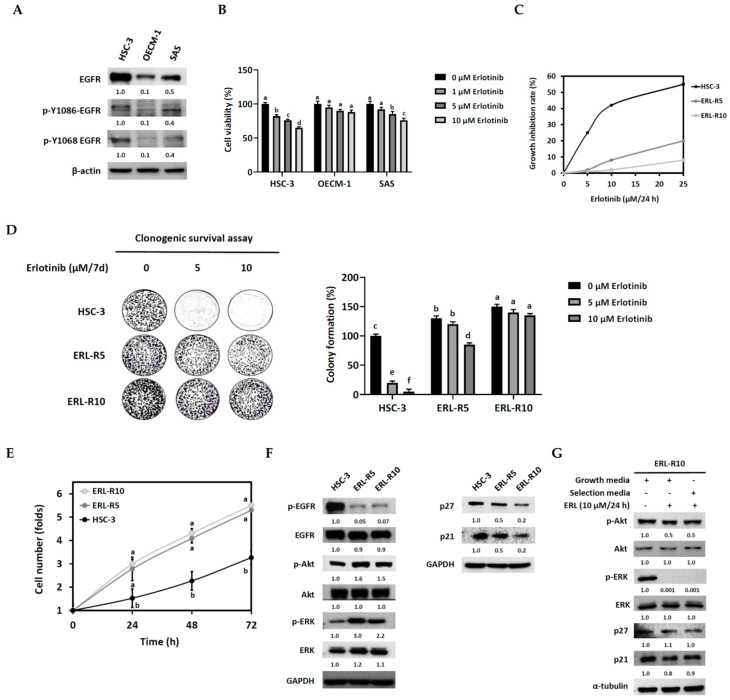
Establishment of erlotinib-resistant oral cancer cell lines. HSC-3, OECM-1, and SAS cells were cultured to 80% confluence, and (**A**) cell lysates were subjected to Western blot analysis to detect the protein levels of EGFR and p-EGFR. β-actin served as an internal control, and each protein expression level was quantified as a relative number to that of HSC-3 cells. (**B**) HSC-3, OECM-1, and SAS cells were cultured with or without erlotinib for 24 h, followed by an MTT cell viability assay. Each bar represented a cell number as a percentage of the control (0 μM erlotinib). Values are mean ± SD, *n* = 3. ^abcd^ Values without the same letter differ significantly, *p* < 0.05. (**C**–**G**) Assessment of erlotinib-resistant cell lines. (**C**) HSC-3, ERL-R5, and ERL-R10 cells were cultured with erlotinib (0, 5, 10, 25 μM) for 24 h and viable cell number was determined by an MTT assay. Data were expressed as growth inhibition rate relative to the control (0 μM erlotinib). (**D**) HSC-3, ERL-R5, and ERL-R10 cells (2 × 10^3^ cells) were seeded onto a 6-well plate and cultured with erlotinib (0, 5, 10 μM) for 7 days and stained with 0.5% crystal violate. Each bar represented colony formation as a percentage of the control (0 μM erlotinib). Values are mean ± SD, *n* = 3. ^abcdef^ Values with different letter differ significantly, *p* < 0.05. (**E**) HSC-3, ERL-R5, and ERL-R10 cells (1.5 × 10^4^ cells) were cultured in 3-cm dishes for 24 h and viable cell number was determined by an MTT assay. Values are mean ± SD, *n* = 3. ^ab^ Values with different letter differ significantly at indicated time point, *p* < 0.05. (**F**) Cell lysates from HSC-3, ERL-R5, and ERL-R10 cells with 80% confluence were subjected to Western blot analysis for indicated proteins. (**G**) ERL-10 cells were cultured in growth media with or without erlotinib or in selection media to 80% confluence, and cell lysates were collected to determine indicated proteins by Western blot analysis. Each protein band was quantified as a relative number to that of ERL-10 cells in growth media.

**Figure 2 cells-12-00179-f002:**
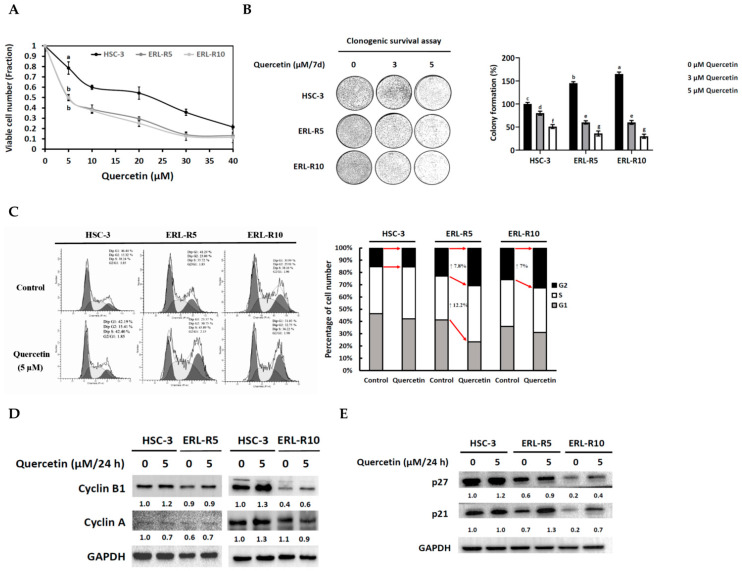
Erlotinib-resistant OSCC cells are sensitive to the growth-inhibitory efficacy of quercetin. (**A**) After seeded in a 6-well plate, HSC-3, ERL-R5, and ERL-R10 cells were treated with quercetin (0~40 μM) for 72 h and viable cell number was determined by a trypan blue staining assay. Values are mean ± SD, *n* = 3. ^ab^ Values with different letter differ significantly at indicated time point, *p* < 0.05. (**B**) HSC-3, ERL-R5, and ERL-R10 cells (2 × 10^3^ cells) were seeded onto a 6-well plate and cultured with quercetin (0, 3, 5 μM) for 7 days. Colony formation was quantified as a percentage of the control (0 μM quercetin). Values are mean ± SD, *n* = 3. ^abcdefg^ Values with different letter differ significantly, *p* < 0.05. (**C**) HSC-3, ERL-R5, and ERL-R10 cells were cultured with the addition of quercetin (0, 5 μM/24 h) and subjected to cell cycle analysis using a FACScan flow cytometer. (**D**,**E**) Cell lysates from HSC-3, ERL-R5, and ERL-R10 cells cultured with or without adding 5 μM quercetin for 24 h were subjected to Western blot analysis for indicated proteins.

**Figure 3 cells-12-00179-f003:**
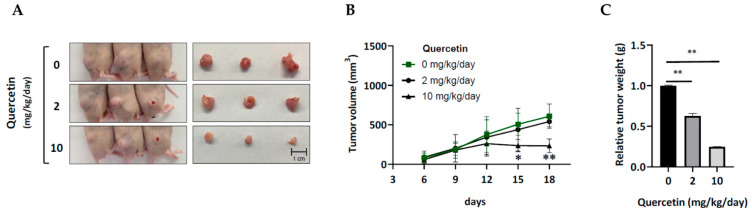
Quercetin suppresses tumor growth of erlotinib-resistant HSC-3 cells in vivo. ERL-R5 cells were subcutaneously injected into the dorsal flank of male nude mice, and quercetin (0, 2, 10 mg/kg/day) was administrated with intraperitoneal injection (*n* = 6 per group). (**A**) The images represented the tumor volume and tumor size in control and two quercetin-treated groups before and after mice were sacrificed, respectively. (**B**) The growth curve of tumor volumes was examined and (**C**) tumors were excised and weighed at the end of the experiment. A two-way ANOVA followed by Tukey’s test was used to assess the differences of tumor volume and weight between the control and quercetin groups (* *p* < 0.05, ** *p* < 0.01).

**Figure 4 cells-12-00179-f004:**
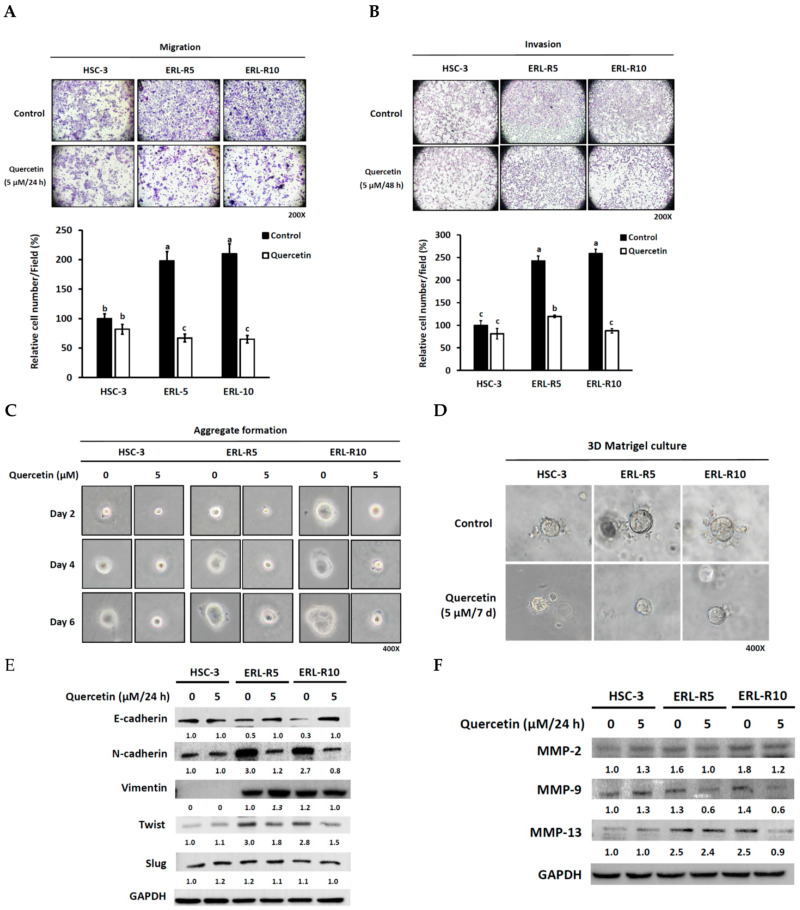
The aggressive phenotype of erlotinib-resistant OSCC cells is sensitive to quercetin. (**A**–**D**) Impact of quercetin on cell (**A**) migration, (**B**) invasion, (**C**) aggregation, and (**D**) sphere formation of erlotinib-resistant cells. HSC-3, ERL-R5, and ERL-R10 cells were plated in the upper chamber in a serum-free medium with or without quercetin (5 μM) and allowed to (**A**) migrate or (**B**) invade for 24 h with the addition of a chemoattractant (10% fetal bovine serum-containing medium) in the lower chamber. Cells that migrated or invaded to the other side of membrane were photographed under 200X magnification and counted in three randomly selected microscopic fields. Values are mean ± SD, *n* = 3. ^abc^ Values with different letter differ significantly at indicated time point, *p* < 0.05. (**C**) HSC-3, ERL-R5, and ERL-R10 (2 × 10^3^ cells) were seeded into low adhesive 35-mm culture dishes and cultured in regular growth media with or without 5 μM quercetin at 37 °C under 5% (*v*/*v*) CO_2_. The aggregation formation was observed on the indicated day and recorded by a phase-contrast microscope. (**D**) HSC-3, ERL-R5, and ERL-R10 cells were grown with or without the addition of 5 μM quercetin in a 3D Matrigel culture for 7 days. The arrows indicate cells invading the surrounding Matrigel. The pictures were taken under 400 magnifications. (**E**,**F**) After 24 h of exposure to quercetin (5 μM) or not, cell lysates from HSC-3, ERL-R5, and ERL-R10 cells were collected and subjected to Western blotting for the detection of (**E**) E-cadherin, N-cadherin, Vimentin, Twist, and Slug, and (**F**) MMP-2, MMP-9, and MMP-13 protein levels.

**Figure 5 cells-12-00179-f005:**
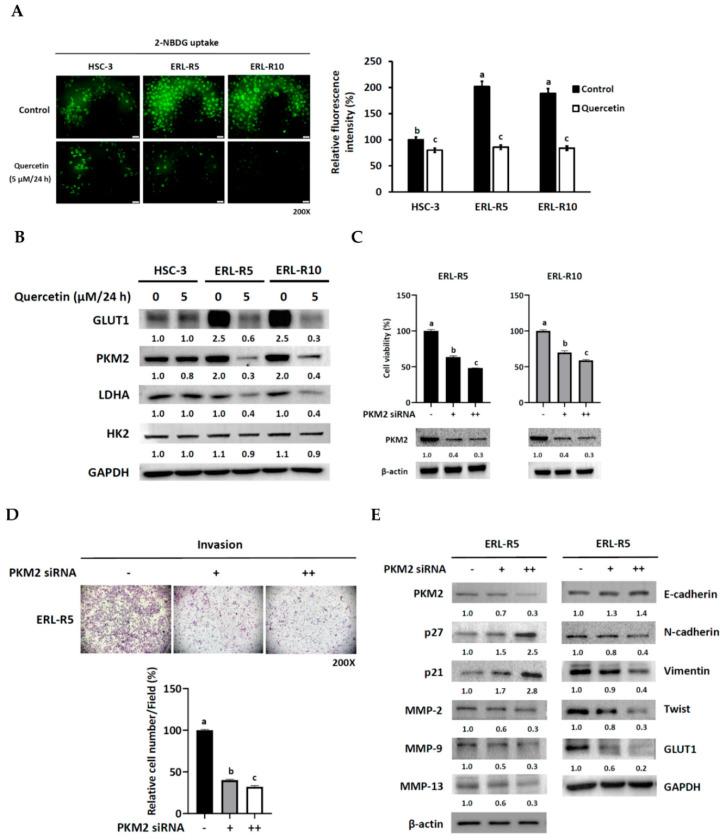
Quercetin modulates glucose utilization and PKM2 expression in erlotinib-resistant cells. (**A**) HSC-3, ERL-R5, and ERL-R10 cells (2 × 10^4^/24-well plate) were cultured in media containing quercetin (5 μM) or not for 24 h after seeded, followed by low glucose media with 2-NBDG for 10 min. Photographs were taken by a fluorescent microscope, and quantified as relative fluorescence intensity to the control HSC-3 with no quercetin. (**B**) After 24 h of treatment with or without quercetin (5 μM), cell lysates from HSC-3, ERL-R5, and ERL-R10 cells were collected and subjected to Western blotting for the detection of GLUT1, PKM2, HK2, and LDHA protein levels. GAPDH served as a loading control. (**C**–**E**) PKM2 knockdown mimics the quercetin effect. Erlotinib-resistant HSC-3 cells were transfected with PKM2 siRNA (0, 10, 20 μM) for 24 h and then subjected to determination of (**C**) cell viability by an MTT assay, (**D**) invading capacity by a Matrigel invasion assay, and (**E**) their related protein levels by Western blot analysis. Values are mean ± SD, *n* = 3. ^abc^ Values with different letter differ significantly at indicated time point, *p* < 0.05.

**Figure 6 cells-12-00179-f006:**
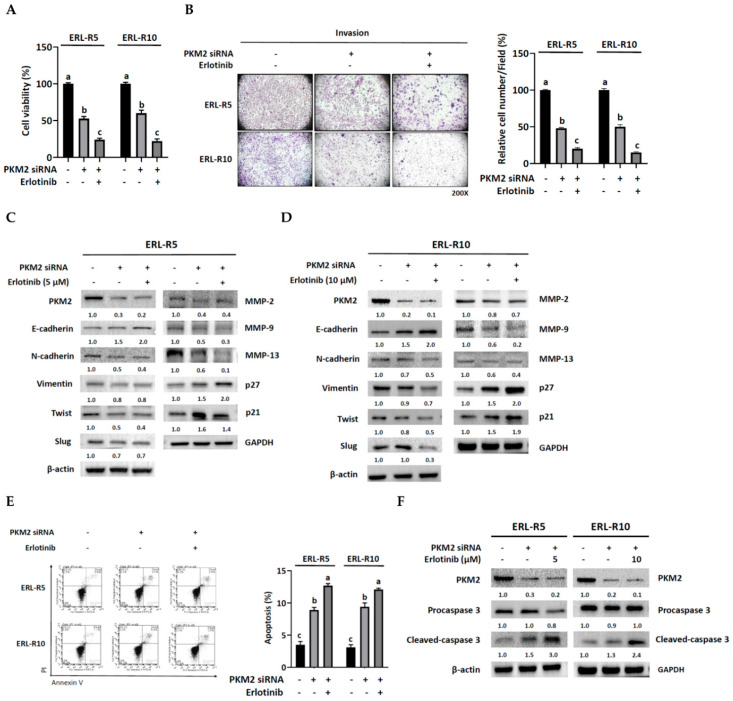
PKM2 knockdown re-sensitizes resistant cells to erlotinib. Both ERL-R5 and ERL-R10 resistant cells transfected with or without PKM2 siRNA were treated with indicated concentration of erlotinib for 24 h, and then cells were subjected to (**A**,**B**) Western blot analysis for indicated proteins, (**C**) an MTT cell viability assay, (**D**) a Matrigel invasion assay, and (**E**) a flowcytometric analysis for cell apoptosis and (**F**) Western blot analysis for cleaved-caspase 3 protein expression. Values are mean ± SD, *n* = 3. ^abc^ Values with different letter differ significantly at indicated time point, *p* < 0.05.

**Figure 7 cells-12-00179-f007:**
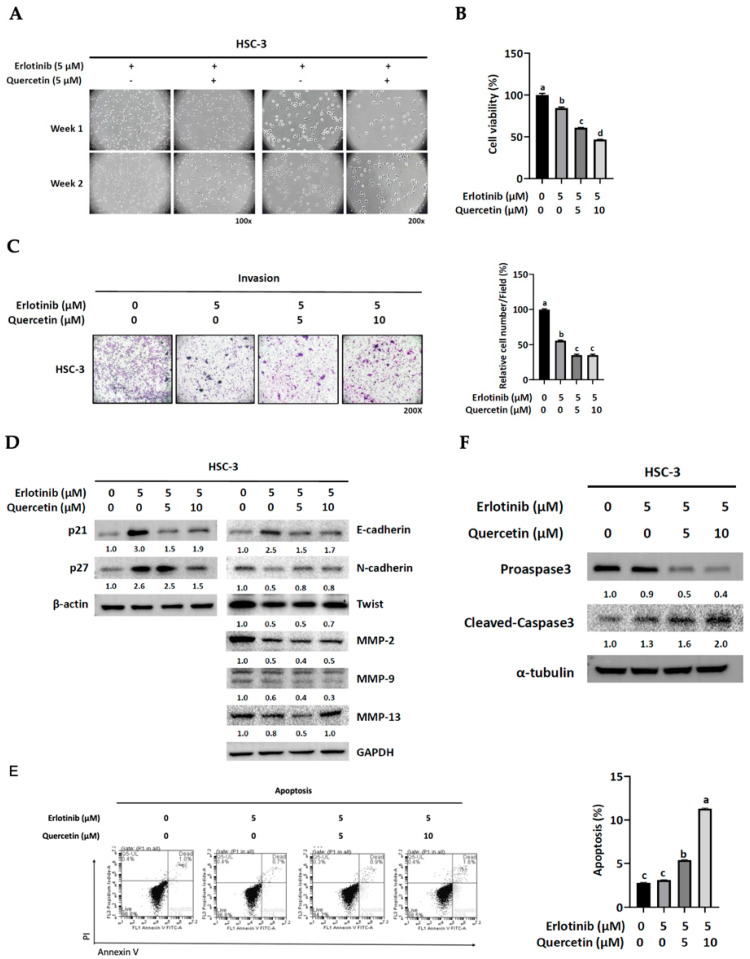
Effect of quercetin on the development of acquired resistance to erlotinib. (**A**) HSC-3 cells were cultured in media containing erlotinib (5 μM) with or without quercetin (5 μM) for 2 weeks and photographed under a phase-contrast microscope. (**B**–**F**) HSC-3 cells were cultured in media containing erlotinib (5 μM) with quercetin (0, 5, 10 μM) for 24 h and then subjected to (**B**) an MTT cell viability, (**C**) a Matrigel invasion assay, (**D**) Western blot analysis for protein levels of p21, p27, E-cadherin, N-cadherin, Twist, MMP-2, -9, and -13, and (**E**) a flowcytometric analysis for cell apoptosis and (**F**) Western blot analysis for cleaved-caspase 3 protein expression. Values are mean ± SD, *n* = 3. ^abcd^ Values with different letter differ significantly at indicated time point, *p* < 0.05.

**Figure 8 cells-12-00179-f008:**
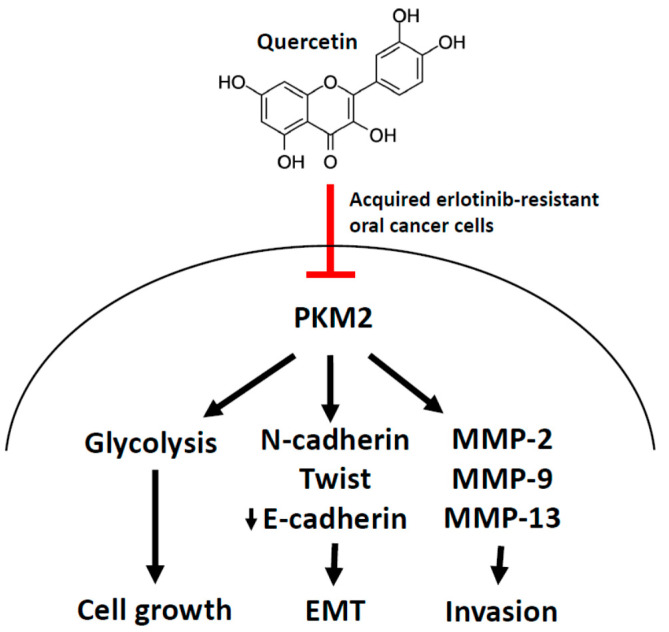
A schematic model depicting the role of quercetin and PKM2 in the regulation of erlotinib-resistant oral cancer cells. PKM2 was up-regulated in oral cancer cells resistant to erlotinib, which was highly associated with elevated glucose utilization, growth rate, EMT, and invasive progression of the erlotinib-resistant cells. Quercetin at 5 µM potently hindered these acquired aggressiveness features in resistant cells by suppressing the PKM2 expression.

## Data Availability

The data that support the findings of this study are available from the corresponding author upon reasonable request.

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
