# Peer review of "Oral Squamous Cell Carcinoma Cells with Acquired Resistance to Erlotinib Are Sensitive to Anti-Cancer Effect of Quercetin via Pyruvate Kinase M2 (PKM2)"

_cells, 2023, doi:10.3390/cells12010179_

Round 1

Reviewer 1 Report

This is an excellent study by Chan et al, where the authors have shown the anti-tumor efficacy of quercetin in erlotinib resistant OSCC line (HSC-3). In vitro studies suggest that quercetin reduces cell viability, induces cell cycle arrest and blocks EMT in drug resistant lines. Quercetin also blocks the glucose uptake in drug resistant cells. Finally, the in vivo xenograft studies suggests that Quercetin suppresses tumor growth of erlotinib-resistant HSC-3. Overall, the finding is novel and very interesting. But poor experimental design severely impacts the importance of the study. Here are my major issues

·         The authors are requested to kindly remove “very low concentration” from abstract and other part of the stories. In an in vitro set up, 5 micromole is already a high concentrations. A molecule which elicits its biological effect in nanomole has a lot of potential in clinical translation.

·         This study is mostly focused on evaluating the anti-tumor  efficacy of Quercetin as a single agent  in erlotinib resistant lines. Only in figure 7, the efficacy of combination (Quercetin amdnd erlotinib) has been evaluated. This part should be expanded. Like different doses of this combination should be tried, i.e. starting from nanomole to micromole concentration of  both Quercetin amdnd erlotinib. Here authors may obtained desired result at a very low concentration of drugs. The combination index should be determined, which will depict if this combination is either additive or synergistic.  Ideally the xenograft should be done in suitable dosage of combination.   

Reviewer 2 Report

In the present study, the authors demonstrated that quercetin at 5 μM significantly caused 464 cell growth arrest, reduced glucose utilization, and blunted invasive progression in erlo- 465 tinib-resistant HSC-3 OSCC cells. The study design is appropriate and conclusion drawn supports the results obtained in the present study. However, some minor modifications are needed, which are as follows:

- Add a statement about what is quercetin in abstract.

- At many places in the methodology section, the authors mentioned that " Procedure was described previously". It is recommended that the details of the procedure should be provided as supplementary material.

- Kindly elaborate on the limitations and future prospects at appropriate place in the discussion.

Round 2

Reviewer 1 Report

Authors have not addressed the concern-raised up to the satisfaction.  It is important to determine the combination index of quercetin and erlotinib in erlotinib-resistant HSC-3 cells. This will determine if this combination is additive or synergestic.